# Effects of trauma-related amputations in children on caregivers: An exploratory descriptive study in a developing country

**Patience Achiamaa Barnie** [1,2] *, **Veronica Millicent Dzomeku**[1], **Abigail Aban Tetteh**[1], **Bernard Asamoah Barnie**[3], **Charles Mock**[4], **Peter Donkor**[5]

**1** School of Nursing and Midwifery, Kwame Nkrumah University of Science and Technology, Kumasi, Ghana, **2** Nursing and Midwifery Training College, Fomena, Adansi, Ghana, **3** School of Public Health, Kwame Nkrumah University of Science and Technology, Kumasi, Ghana, **4** Department of Surgery, University of Washington, Seattle, WA, United States of America, **5** Department of Surgery, Kwame Nkrumah University of Science and Technology, Kumasi, Ghana

* pattyboat19@gmail.com

## Abstract

Amputation in children is rare. However, in recent times, amputation in children has increased and trauma is the leading cause in Ghana. Few studies on the effects of amputation on caregivers particularly of children are available. This study aimed to explore the effects of trauma-related amputations in children on caregivers using qualitative descriptive phenomenological approach. In-depth interviews were conducted with semi-structured interview guide. Ten (10) informal caregivers were purposively selected from the trauma registry of a tertiary facility in Ghana. Data were analyzed manually using the thematic approach described by Collaizi. The findings revealed that trauma-related amputations in children affect the work-role, social life, finances and mental health of the caregivers. Provision of counselling services to address the mental health needs of caregivers and decentralization of orthopaedic and rehabilitation services would lessen the burden of caregiving.

## Introduction

Amputation is the removal of a body part such as a finger, toe, hand, foot, or arm. Amputation may be necessary in certain cases due to injuries, infections, tumors, congenital defects, or vascular anomalies [1]. Globally about 57.7million people live with limb amputation due to traumatic causes [2]. The USA records 1.6million people are living with a loss of limb and the number is projected to double by the year 2050 [3]. In Ghana, the number of estimated amputees, which stands at 248,299, is on the rise annually owing to various factors such as an increase in the number of road traffic crashes [4].

The epidemiology of trauma-related amputations in children is not well documented in Ghana, as well as in most developing countries. The only study conducted in Ghana recorded thirty-four (34) trauma-related amputations among children in one treatment center over 3.5years [5]. This finding indicated a remarkable prevalence in children. Trauma is known to be the leading cause of amputation among children in Ghana [5].

**Funding:** This study was supported by grant D43 TW007267 from the Fogarty International Center at the US National Institutes of Health (NIH). AUTHORS: CM and PD The funders Charles Mock (CM) and Peter Donkor (PD) played a role in the study. CM was involved in analysis, review and decision to publish the manuscript PD was involved in review of manuscript, supervision, decision to publish manuscript and project administration.

**Competing interests:** The authors of this manuscript have no competing interest.

When amputation occurs, caregivers are the most affected. Caregivers of amputees experience a wide range of unpleasant feelings. It can be a horrific and life-altering experience that leaves the victim and their caregivers unable to function normally and become dependent on others [6]. The trauma and stress experienced by caregivers increase significantly when children are involved, considering the vulnerability of children. It can also be difficult for caregivers of amputee children to adjust to their new life, which has an impact on many facets of their lives [5–7].

The Paradigm Model of the Experiences of Family Carers of Patients was adapted to guide this study [8]. The model explores the complexities of caregiving, considering both positive and negative aspects. It emphasizes how contextual factors, such as service quality and financial situations, and caring circumstances shape caregiver activities and dynamics.

Quality of service encompasses healthcare quality and administrative procedures. The financial circumstances concern how caregivers manage expenses related to their patient's care. These are the additional fees levied in addition to the cost covered by national health insurance. Finally, caregiving circumstances are the concerns that surround the patient's care. It explains the many disputes between the patient and the caregiver. Intervening conditions impact the phenomenon directly or indirectly. The caregiving process and societal support play roles in influencing the phenomenon. The model proposed by [8] defines caregivers' responses to challenges as actions. Caregivers adapt by employing coping techniques, sharing care responsibilities, and making compromises. The consequences in the model are the physical, psychological, economic, and social impacts experienced by caregivers due to caring and the process of caring. The model guided the objectives of the study.

Although a lot of studies have been done regarding amputation, they used quantitative approaches and reported on the pattern of amputation in both children and adults [4, 5, 9, 10], the challenges of adult amputees, and the problems of amputation after surgery [11]. Other authors also focused on the availability of prosthesis to promote the quality of life post-amputation [1, 6, 12]. A single qualitative study was carried out, focusing on negative emotional experiences of amputees and their caregivers in Ghana. However, the study specifically concentrated on caregivers of adult amputees [6]. Since there is limited knowledge on trauma-related amputations in children and specific effects on the caregivers, this study sought to explore the effects of trauma-induced amputations in children on their caregivers, and thus illuminate different facets of the caregivers' lives. Gaining insight into the caregivers' experiences will guide policies addressing the healthcare requirements of amputees and their support networks.

## Materials and methods

### Design

The study was a qualitative one, and a descriptive phenomenological approach was used to explore this phenomenon. The approach was appropriate because it was used to explore the lived experiences of different people about the phenomena with very little or no attempt to interpret them [13, 14]. The approach is also useful for investigating different opinions of human beings and their interpretation of lived experiences in their regular state [15]. The design was appropriate for this study since little information is known on the effects of trauma-related amputations in children. Information gathered from a qualitative descriptive study are extensive descriptions of participants which are mostly expressed in words, this information provides better understanding to the phenomenon understudy [14]. This approach is useful, in the Ghanaian setting where there is paucity of literature on trauma-related amputations among children.

## Study setting and site

Our study was conducted in a tertiary facility in Ghana, Komfo Anokye Teaching Hospital, one of the five teaching hospitals in Ghana, which It is located in Kumasi, Ashanti Region of Ghana. It serves as the major referral center accessible to all regions that share a border with Ashanti Region of Ghana. The Trauma and Orthopaedic Directorate of the Komfo Anokye Teaching Hospital (KATH) was the source for the recruitments of the participants used for the study. Information on caregivers was taken from the trauma registry of the directorate.

## Population

The target population for this study was the immediate informal caregivers of children who had undergone amputations with traumatic indications at the Trauma and Orthopedic Directorate of a tertiary facility in Kumasi, Ashanti Region of Ghana. Informal caregivers are family caregivers who are offering unpaid services and taking care of amputee children

## Inclusion and exclusion criteria

The selection of participants was based on the following inclusion criteria. 1) Caregivers of children below 18years who had undergone amputations due to traumatic indications. 2) Caregivers whose children had been discharged from the facility. 3) Caregivers who gave consent to take part in the study. 4) Caregivers who were aged 18years and above.

Caregivers of children with trauma-related amputations who had been discharged for less than 2 months as well as caregivers of children with amputation who were still on admission were excluded from this study. These caregivers were excluded because they might not have had enough exposure and experience to share.

## Sample size and sampling

Purposive sampling technique was used to select caregivers of children who had undergone amputations due to trauma. The children of these caregivers had received treatment at the tertiary facility and discharged home. The researcher purposively requested from the trauma registry details of children below 18years who had undergone amputation due to traumatic indication from the year 2013 to the year 2022 at the facility. After the 10th interview, the data reached saturation. Data saturation is reached in qualitative study when enough information has been gathered in a study and no new information emerges during coding and analysis of data [16, 17].

## Data collection instrument

A semi-structured interview guide was used to collect the data. This guide was formulated in alignment with the study's objectives and the pertinent literature on the subject matter. The interview guide had two (2) main sections. Section A focused on the demographic and professional background of participants. Section B had three (3) main questions that focused on information that expressed the effects of amputation on the caregiver.

## Data collection procedure

Prior to commencing the study, a representative from the trauma registry was introduced to us, by the Head of Trauma and Orthopedic Directorate on the 13th of April, 2023. Data on children who had undergone amputation between the year 2013 to 2022 was retrieved. All children with amputations, as a result of trauma were selected. The names of the children and the address and phone numbers of caregivers were received and contacted for the study. The data

collection tool was piloted with two similar participants in the same facility. This validation process aimed to ascertain that the tool effectively prompted the requisite inquiries to solicit responses pertinent to the research inquiries.

Data was gathered from caregivers from the 24th April, to 20th May, 2023. Interviews were conducted using a semi-structured interview guide at a convenient place and time of each participant. Eight (8) face-to-face interviews were conducted in the homes of the caregivers, while two (2) were conducted at the facility. Interviews that were conducted at the facility, took place in an office at the disaster block of the facility. The researcher explained the aim of the study to caregivers and informed them that participation was entirely voluntary and consent was sought for participation and the recording of interviews. Participants were notified in the consent form that, despite signing it, they retained the option to withdraw from participation at any point. Additionally, they were informed that the collected data would solely serve the study's purposes. Furthermore, they were made aware that only the researcher and her team would have access to the data. On average, each interview lasted 40 minutes, with variations in duration attributed to factors such as the child's age and the impact of the experience on the caregiver. For each interview, the local dialect (Twi) was employed since the majority of participants were more proficient in Twi. Permission was obtained from participants to tape-record the interviews, which were subsequently translated into English. Throughout the interviews, the researcher also documented non-verbal expressions in a field notebook. Following each interview, the first author transcribed all audiotapes word for word for further analysis.

## Ethical considerations

The Institutional Review Board of Komfo Anokye Teaching Hospital gave ethical approval with approval number KATH IRB/AP/13/23 for the conduct of this study. Approval was also given by the ethical Committee on Human Research, Publication and Ethics (CHRPE) of Kwame Nkrumah University of Science and Technology (CHRPE/AP/219/23). Official permission was sought from the Trauma and Orthopedics Directorate of Komfo Anokye Teaching Hospital and the Head of the Trauma Registry with an introductory letter from the School of Nursing, College of Health, Kwame Nkrumah University of Science and Technology. Participants were approached to obtain their informed consent, with clear explanations provided regarding the study's purpose, objectives, and potential risks and benefits. Participant records were securely stored and accessible only to the researcher and her team, ensuring confidentiality.

## Trustworthiness

In qualitative research, trustworthiness refers to the capacity to ascertain whether the researcher's conclusions accurately reflect the experiences of the participants [18]. Confirmability, credibility, transferability, and dependability are the key elements of trustworthiness in qualitative studies [18].

To maintain credibility, the researcher enlisted participants who met specific inclusion criteria and could offer a diverse range of information on the phenomenon. Following each interview, member checks were conducted, allowing the researcher to validate participants' responses before drawing conclusions from the data. Gathering feedback from participants and consolidating it was crucial in accurately and clearly presenting their narratives. Additionally, each interview was transcribed verbatim and coded on the same day it was conducted, aiding the researcher in correlating responses with the study's objectives. Dependability/Reliability refers to whether or not the study can be duplicated by other researchers [19]. In order to ensure reliability, the researcher provided detailed description of the research design and

the procedures used for collecting data, as well as analysing data in the final report. The research methodology was extensively described as it was applied to the study. Additionally, in ensuring reliability, the same interview guide, tape recorder, and analysis process were used for the interviews or data collected from all participants in the study. Field documentation, including field notes and verbal and nonverbal exchanges, were reviewed with the supervisor by the researcher as well.

Confirmability in qualitative study is the ability of the researcher to be impartial. Unpacking personal bias can be accomplished by bracketing interview. To ensure confirmability, the researcher made sure all conclusions made from the study were based on the exact experiences and thoughts of the participants and not the opinions of the researcher. A clear coding schema was used to identify codes and patterns [18].

Transferability is the extent to which findings from a study can be applied in other settings [20]. The researcher ensured detailed descriptions of research setting, methodology and background of the participants used for the study. Transcripts and analysed documents were kept for purpose of references. This will make it easy for other researchers to extend conclusions from this study unto other similar studies.

### Data analysis

All audios were transcribed verbatim immediately after every interview by the lead author and a trained research assistant. The transcripts were double checked by PAB and AAT. Thematic analysis technique was used to analyse the interview data manually simultaneously. The analysis involved seven phases: Researchers, PAB and VMD read through the transcripts many times to familiarize with the findings. All transcripts were read line by line, and some significant statements were noted. A coding strategy developed was by PAB and BAB to formulate meaning to responses to the interview questions. Differences in the themes were resolved by involving another researcher, PD.

Identified codes were clustered into themes and subthemes based on the research objectives and the paradigm model of the experiences of family carers of patients providing care. They were later fine-tuned with continuous analysis by PAB, VMD and BAB. Finally, engaging quotes were selected and linked with the research topic and literature discussed in this study. Only participants who met the inclusion criteria were recruited, which helped the researcher maintain the study's credibility. Also, to ensure that the interpreted data matched exactly what had been communicated during the interviews, PAB, AAT, made sure each participant received feedback from the interviews that confirmed what participants had said during each interview conducted.

## Results

### Participants

A sample of 10caregivers were recruited into the study. Participants were aged between 21-58years. Participants were made of five (5) females who were mothers, four (4) males who were fathers and uncles. Nine (9) of the participants had informal employment, whereas one (1) had formal employment. Table 1 below is a summary of the background characteristics of participants of the study.

### Main findings

Three major themes that emerged from the study were daily activities/social life, finances, and mental health and emotions. The major themes and their subthemes are summarized in the Table 2 below.

**Table 1. Background characteristics of participants.**

| S/N | Age | Marital status | No. of children | Occupation | Level of education | Religion | Relationship to child | Period of caregiving | Age of child | Mechanism of injury | Limb involved |
|---|---|---|---|---|---|---|---|---|---|---|---|
| 001 | 47 | Married | 4 | Trader | None | Muslim | Mother | 2 years | 15 | Gunshot | Lower limb (above knee) |
| 002 | 40 | Married | 3 | Farmer | Junior High School | Christian | Mother | 1 year | 8 | Vehicular Knockdown | Lower limb (above knee) |
| 003 | 35 | Widow | 1 | Trader | Junior High School | Christian | Mother | 1 year | 14 | Vehicular Knockdown | Lower limb (above knee) |
| 004 | 58 | Married | 5 | Farmer | Primary school | Christian | Uncle | 10 months | 6 | Vehicular Knockdown | Lower limb (foot) |
| 005 | 21 | Single mother | 1 | Hairdresser | Senior High School | Christian | Mother | 1 year | 4 | Child play Accident | Foot (toe) |
| 006 | 45 | Married | 3 | Farmer | Middle school | Christian | Uncle | 2 year | 14 | Vehicular Knockdown | Lower limb (above knee) |
| 007 | 44 | Divorced | 4 | Trader | Junior High School | Christian | Mother | 10 years | 11 | Vehicular Knockdown | Lower limb (above knee) |
| 008 | 70 | Married | 8 | Corn mill operator | None | Muslim | Father | 2 years | 6 | Child play Accident | Lower limb (below knee) |
| 009 | 40 | Married | 6 | Miner | Vocational Training | Christian | Father | 2 years | 13 | Iatrogenic-related Trauma | Both limbs (above knee) |
| 010 | 42 | Married | 1 | Farmer | Middle school | Christian | Father | 3 years | 15 | Gunshot | Lower limb (above knee) |

Source: Field Data, 2023

**Daily life activities/social life.** This theme explores how caregivers' daily routines are impacted by caring for their amputated children. The underlying sub-themes under this theme were: (1) restricted movement (2) exhaustion

*Restricted movement.* Participants in this study expressed the feeling of constrain in their movements and unable to leave their children for extended periods due to caregiving responsibilities. Participants cited that they had to cease their usual activities and prioritize caring for their children. This entailed being constantly available to provide support, leaving them unable to go anywhere else. This is illustrated with the quotes below:

*". . .now I'm not able to go anywhere unlike before when he had both legs. . . my work has been disrupted. . . there has been changes. . . my mother passed on yesterday and I was*

**Table 2. Themes and Sub-themes generated.**

| Themes | Sub- themes |
|---|---|
| *Daily Life Activities/ Social Life* | • Restricted Movement<br>• Exhaustion |
| Finances | • High Cost of Treatment<br>• Opportunity Cost (Earning loss) |
| Mental Health and Emotions | • Feeling of guilt and sadness<br>• Coping mechanisms |

*supposed to go but I could not go because of [name of son]"* **(P003_mother of child, lower limb above knee)**

*"...the child is not really that matured, that's why I'm not able to leave him and go where you have to go because when he needs assistance, someone will have to be there so we always have to be around him."* **(P006_uncle of child, lower limb above knee)**

*"Oh, he had a sibling who was in school at the time and stopped schooling to cater for him... so our movement has really been affected..."* **(P010_father of child, lower limb above knee)**

*Exhaustion.* Caregivers expressed the stress that comes with taking care of the children with amputation. They expressed the exhaustion from the physical strength that goes into lifting and supporting the children in the performance of the various activities of daily living. This is shown in the following narratives:

*"because he is young, we are the ones who do everything for him. It's only recently that he is able to sweep his room"* **(P001_mother of child, lower limb above knee)**

*"I was the one carrying her around all the time for about 4 months..."* **(P002_mother of child, lower limb above knee)**

*"...now if she has to do anything, we have to do it for her. It gets tiring..."* **(P009_father of child, both limbs above knee)**

**Finances.** This theme explores how the participants are affected financially due to the amputation of their children. This theme again describes the financial support involved in treatment. Two sub-themes emerged under this theme: (1) high cost of treatment (2) opportunity cost (earning loss).

*High cost of treatment.* Participants also narrated that they provided financial support to cover for treatment cost since there is limited coverage of treatment by the National Health Insurance Scheme (NHIS). The Ghana NHIS provides partial coverage for the treatment cost of amputation. This results in caregivers paying for almost all of the cost incurred throughout the process of receiving healthcare. These are shared in quotes below:

*"You know when you go to the hospital, everything involves money so we were challenged a bit there... and life in the rural area is not easy. It is even difficult to sell for people to buy so we do not really have money and this also happened."* **(P006_uncle of child, lower limb above knee)**

Some participants shared how they had borrowed from people and they had still not been able to pay their debt. They described the cost involved in attending regular hospital reviews. This cost is due to the long distance they have to travel to access orthopedic and rehabilitation services. This hospital reviews are for a period of at least three times in a week for review. Below are some quotes that illustrate it:

*"We have been going to the hospital every 2 days since he was discharged..."* **(P003_mother of child, lower limb above knee)**

*"...send her to the hospital at Asafo every 3 days and it's been 10 months now..."*
**(P004_uncle of child, lower limb foot)**

*"In terms of finances... the frequent visitation to the hospital has made things difficult a bit because now we go there every month... so the finance is not so good at all."* **(P002_mother of child, lower limb above knee)**

*Opportunity cost (Earning loss).* Participants described the financial losses they incurred indirectly due the event of amputation of their children.

Their stories highlight how they had to leave their jobs and stay in the hospital and be beside their children during and after the amputation. These narratives shed light on the financial hardships faced by participants. Since most of the participants had no formal employment, the entire period of being absent from work, led to massive loss of income. These financial dilemmas are shared in the quotes below:

*"We have really been affected financially... most of our works were affected because I had to be with all the time at Gee* [Komfo Anokye Teaching Hospital] *... we spent everything we had, my wife used to trade, we spent all the capital and went into borrowing... in fact, we are not even done with the loans we took from people."* **(P010_father of child, lower limb above knee)**

*"Her mother was operating a chop bar so she had some amount of money but since the child got injured, she has been sending her to Asafo and to this place as well so things are a bit difficult for her now."* **(P004_uncle of child, lower limb foot)**

*"...I am here with my wife and children, my mother had to stop her business at Dunkwa through which she sometimes gave me some support and move here... Secondly, my wife is not able to work because of her condition because there are things that she will need assistance with"* **(P009_father of child, both limbs above knee)**

**Mental health and emotions.** This theme describes the effects of amputation in children on the mental health and emotions of the caregiver. Furthermore, the theme explored some coping mechanisms adopted by caregivers. Three sub-themes were derived from this theme. (1) feeling of guilt and apathy (2) social exclusion (3) comping mechanism.

*Feeling of guilt and sadness.* Majority of the participants made it known that they feel guilty and extremely sad each time the see their children in that situation. They narrated that they sometimes feel guilty for indirectly neglecting their children and they feel responsible for causing the accidents leading to amputation of the children.

Caregivers expressed concerns about their children's future, with some lamenting that their children's condition might prevent them from fully experiencing life. These are shared in the quotes below:

*I was sad... especially seeing him crawl on his knees when he wakes up and wants to go and urinate... also, when I see his mates play football* [speaks crying]*"* **(P007_mother of child, lower limb above knee)**

*"I used to think that because of this, he has delayed or I have delayed in life ..."* **(P010_father of child, lower limb above knee)**

*I'm not able to eat nor sleep, I'm always thinking as to why this problem happened... So, I keep asking why did this happen to me..."* **(P003_mother of child, lower limb above knee**

*"I used to cry a lot... when it happened initially and looking at how my son was just lying down in pain and distress, I could not eat especially the first day they took him to the theatre... I'm not able to eat nor sleep well, I'm always thinking as to why this problem*

*happened. . . So, I keep asking why did this happen to me. . ."* (*P003_mother of child, lower limb above knee*)

*Coping mechanism.* Under this sub-theme, participants shared some coping strategies that they had adopted regardless of the effects of amputation on their lives. The strategies ranged from self-encouragement, community support, and having faith that God knows the best and understands what He does. Their accounts reveal the initial shock, grief, and adjustment period that accompanies such a significant change. However, as they navigate these challenges, they also demonstrate resilience, strength, and the power of faith and support from their communities. This compilation of narratives offers a glimpse into the complex emotional terrain of those affected by limb amputation, highlighting their capacity to find solace and gratitude even in the face of adversity. The quotes below illustrate some of the coping strategies:

*"At first it was tough but other elders also encouraged and advised me and ask me to be grateful because what would I have done if he died. Now even with this condition, both of us are happy even when were alone because the incident has already happened. . ."* **(P003_mother of child, lower limb above knee)**

*"We encourage ourselves because no matter what, the incident has already happened and I cannot give him a new leg so I have to encourage myself although we did not want that but we encourage ourselves so that it would not happen again. We also talk to the child a lot so that he would not consider any negative thought."* **(P006_uncle of child, lower limb above knee)**

## Discussion

This research aimed to investigate how trauma-related amputations in children impact their caregivers. Results revealed that caregivers experience significant effects following the amputation of children, particularly in areas such as their daily routines, finances, and emotional and mental health.

### Daily life activities/social life

Caregivers' daily routines are significantly influenced by their children's conditions, leading to substantial changes in their lifestyles. These effects were experienced with restricted movement of the caregiver and physical exhaustion.

In the Ghanaian settings where there is very limited disability-friendly accommodations, caregivers are confined to their homes to provide constant assistance to their children. Even though in 2006, Ghana passed the National disability Law, Act 715, lack of advocacy, implementation and supervision, these policies on accessibility has been largely overlooked [21]. Many public places in Ghana are built with little or no consideration for persons with disability, therefore caregivers resort to being in their homes with their children all the time. Basic tasks like maintaining personal hygiene become physically demanding, necessitating caregivers to lift their children for bathing. This caregiving responsibility consumes their time, preventing them from engaging in other activities and social events. These are consistent with previous studies that emphasizes the caregiving burden of social strain [22]. Furthermore, caregivers often experience physical exhaustion from their role. When children go through amputation, they become solely dependent on their caregivers. Participants described physical exhaustion from the amount of energy used in assisting their children to perform activities of daily living. This finding was confirmed in several studies from previous authors [22–26]

highlighting the need to promote personal resources and reduce exhaustion in caregivers [22, 25, 26].

## Finances

Finances of caregivers of amputees were identified to have been largely affected by amputation. Amputation of children led to financial loss from high cost of treatment and earning loss of the caregiver.

Participants narrated that they provided financial support to cover for most treatment cost since there is limited coverage of amputation treatment by the National Health Insurance Scheme (NHIS). In Ghana, a significant percentage of hospital expenses is not covered by the National Health Insurance Scheme [27]. This situation puts a lot of stress on the finances of the caregivers of the children. The high cost of treatment involved, cost of medications, cost of procedures and investigations as well as rehabilitation services.

Participants also cited that they also provided financial assistance for other non-treatment costs such as transportation for access to medical care. Additionally, caregivers were burdened to cover long distances with their children for hospital reviews and consultations due to limited orthopedic and rehabilitation services in Ghanaian hospitals thus increasing the financial burden of caregivers. This corroborates the conclusions drawn by [28] regarding the insufficiency of orthopedic specialist services and trauma care in peripheral facilities across Ghana.

Study findings also showed that, children are typically hospitalized for about three months averagely. Considering most participants did not have formal employment, they suffered financial earning losses as they are unable to attend to their daily businesses that fetch them income. This is consistent with findings from [29] highlighting opportunity cost of caregivers. Some caregivers expressed distress over having to dip into their life savings to settle hospital bills, while others resorted to borrowing money. Upon discharge, caregivers could not immediately return to work, and upon doing so, they had to adjust their schedules to accommodate their children's needs, leading to decreased productivity and a reduction in income. This echoes findings from a study [30] regarding the financial impact on parents of children with amputations [28–30].

## Mental health and emotions

Caregivers of amputee children in Ghana experience significant emotional and mental health challenges. The feeling of guilt and sadness was experience by all caregivers. They struggle to comprehend why such circumstances have befallen their families and worry about their children's future and societal integration, especially considering the stigma surrounding disabilities in the country. The discrimination faced by amputee children, particularly among their peers, adds to caregivers' distress, leading them to sometimes cry in private over their children's pain and rejection. Furthermore, the lack of disability-friendly infrastructure in public places like schools and playgrounds exacerbates their concerns about their children's ability to function in society. These worries place a significant psychological strain on caregivers, as observed in various studies, highlighting the importance of addressing the mental health needs of caregivers in similar situations [6, 25]

Intermittent emotional breakdowns manifested as feelings of apathy, sadness, guilt, social exclusion, depression, and disappointment. Some caregivers found themselves blaming their own perceived carelessness and irresponsibility. They expressed regret over not having adequate time to fully accept their children's conditions. This array of negative emotions experienced by caregivers of individuals with disabilities is echoed by other researchers [7, 25, 31].

The study further explored coping mechanism by caregivers which discovered existing coping strategies and adaptive strategies. These strategies included self-encouragement, disengaging from stressful thoughts as well as believing in God. Participants also described community and family support. Our findings correspond with earlier research on coping mechanisms of caregivers [32, 33].

## Strength and limitation of the study

Using the phenomenological approach, the study was able to gain enough insights on the experiences of caregivers. The primary constraint of this research stems from its exclusive reliance on participant reports, which lacked independent verification. Moreover, participants' experiences might have been shaped by additional factors, such as their socio-economic background. Nevertheless, despite these constraints, the results of this study retain significance and offer valuable insights into how trauma-related amputations in children impact their caregivers.

## Implication for research and clinical practice

The findings of this study suggested that trauma-related amputations in children have an enormous effect on the caregivers of these children. Caregivers find themselves obligated to remain behind to take care of their children primarily due to the lack of supportive infrastructure for enabling independence among individuals with disabilities. Consequently, research ought to prioritize proposing policies and establishing infrastructure to enhance the quality of life for people with disabilities and alleviate the caregiving burden. Additionally, cost of healthcare put a toll on the finances of caregivers therefore policies should be made for orthopaedic and rehabilitation services do be decentralized under captured the National Health Insurance Scheme. The study underscores the importance of providing psychological and social assistance to caregivers, alongside addressing their healthcare requirements. Offering counseling services and social support would be advantageous in enhancing the mental well-being of caregivers.

## Conclusion

The current study has illuminated the burden of caregiving on caregivers of children with trauma-related amputations. The inconveniences with caring affect all aspects of the caregivers' lives. The effects were experience on their daily live activities and social life, finances and emotional and mental health. This significant effect on the caregiver highlighted the gaps in healthcare delivery in Ghana. Psychotherapy should be improved since none of the caregivers received any counselling before and after the process of amputation. Also, the National Health Insurance Scheme should be expanded fully to cover cost of treatment of all conditions. Orthopaedics services which are restricted mostly in the tertiary facilities should be decentralized to reduce the cost in accessing healthcare from the peripherals. These measures could improve the burden on caregivers in Ghana.

## Supporting information

**S1 File. Interview guide.**
(DOCX)

**S2 File. Participants' quotes.**
(DOCX)

## Acknowledgments

The authors acknowledge the caregivers and children who participated and shared their experiences with them as well as the Trauma and Orthopedic Directorate for granting permission and providing information on caregivers for the study.

## Author Contributions

**Conceptualization:** Patience Achiamaa Barnie, Veronica Millicent Dzomeku.

**Data curation:** Patience Achiamaa Barnie, Abigail Aban Tetteh, Bernard Asamoah Barnie.

**Formal analysis:** Patience Achiamaa Barnie, Bernard Asamoah Barnie, Charles Mock.

**Funding acquisition:** Charles Mock, Peter Donkor.

**Methodology:** Patience Achiamaa Barnie, Bernard Asamoah Barnie.

**Project administration:** Patience Achiamaa Barnie, Charles Mock, Peter Donkor.

**Supervision:** Veronica Millicent Dzomeku, Charles Mock.

**Validation:** Charles Mock.

**Writing – original draft:** Patience Achiamaa Barnie.

**Writing – review & editing:** Veronica Millicent Dzomeku, Abigail Aban Tetteh, Charles Mock, Peter Donkor.

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
