## [Decision Letter · Decision Letter 0]

16 Sep 2024

PONE-D-24-29492Effects of Trauma-related Amputations in Children on Caregivers: An Exploratory Descriptive Study in a Developing CountryPLOS ONE

Dear Dr. BARNIE,

Thank you for submitting your manuscript to PLOS ONE. After careful consideration, we feel that it has merit but does not fully meet PLOS ONE’s publication criteria as it currently stands. Therefore, we invite you to submit a revised version of the manuscript that addresses the points raised during the review process.

**ACADEMIC EDITOR: **There are some grammatical errors in text. Please revise the whole text using a native academic English editor. Moreover, the authors extracted three themes form the interviews. However, it is not clear what were the number of sub-themes that these themes emerged from them. Please add a table indicating the themes and their related sub-themes. The result and discussion sections should be revised accordingly. Finally, please check the Consolidated criteria for reporting qualitative research (*COREQ*) checklist and revise your text based on its criteria for reporting a qualitative study.

We look forward to receiving your revised manuscript.

Kind regards,

Taher Babaee

Academic Editor

PLOS ONE

Journal Requirements:

1. When submitting your revision, we need you to address these additional requirements. Please ensure that your manuscript meets PLOS ONE's style requirements, including those for file naming. The PLOS ONE style templates can be found at https://journals.plos.org/plosone/s/file?id=wjVg/PLOSOne_formatting_sample_main_body.pdf and https://journals.plos.org/plosone/s/file?id=ba62/PLOSOne_formatting_sample_title_authors_affiliations.pdf 2. Thank you for stating the following financial disclosure: "This study was supported by grant D43 TW007267 from the Fogarty International Center at the US National Institutes of Health (NIH).   AUTHORS: CM and PD".  Please state what role the funders took in the study.  If the funders had no role, please state: "The funders had no role in study design, data collection and analysis, decision to publish, or preparation of the manuscript." If this statement is not correct you must amend it as needed. Please include this amended Role of Funder statement in your cover letter; we will change the online submission form on your behalf. 3. We note that your Data Availability Statement is currently as follows: "All relevant data are within the manuscript and its Supporting Information files". Please confirm at this time whether or not your submission contains all raw data required to replicate the results of your study. Authors must share the “minimal data set” for their submission. PLOS defines the minimal data set to consist of the data required to replicate all study findings reported in the article, as well as related metadata and methods (https://journals.plos.org/plosone/s/data-availability#loc-minimal-data-set-definition). For example, authors should submit the following data: - The values behind the means, standard deviations and other measures reported;- The values used to build graphs;- The points extracted from images for analysis. Authors do not need to submit their entire data set if only a portion of the data was used in the reported study. If your submission does not contain these data, please either upload them as Supporting Information files or deposit them to a stable, public repository and provide us with the relevant URLs, DOIs, or accession numbers. For a list of recommended repositories, please see https://journals.plos.org/plosone/s/recommended-repositories. If there are ethical or legal restrictions on sharing a de-identified data set, please explain them in detail (e.g., data contain potentially sensitive information, data are owned by a third-party organization, etc.) and who has imposed them (e.g., an ethics committee). Please also provide contact information for a data access committee, ethics committee, or other institutional body to which data requests may be sent. If data are owned by a third party, please indicate how others may request data access. 4. When completing the data availability statement of the submission form, you indicated that you will make your data available on acceptance. We strongly recommend all authors decide on a data sharing plan before acceptance, as the process can be lengthy and hold up publication timelines. Please note that, though access restrictions are acceptable now, your entire data will need to be made freely accessible if your manuscript is accepted for publication. This policy applies to all data except where public deposition would breach compliance with the protocol approved by your research ethics board. If you are unable to adhere to our open data policy, please kindly revise your statement to explain your reasoning and we will seek the editor's input on an exemption. Please be assured that, once you have provided your new statement, the assessment of your exemption will not hold up the peer review process. 5. Your ethics statement should only appear in the Methods section of your manuscript. If your ethics statement is written in any section besides the Methods, please move it to the Methods section and delete it from any other section. Please ensure that your ethics statement is included in your manuscript, as the ethics statement entered into the online submission form will not be published alongside your manuscript.

Reviewers' comments:

Reviewer's Responses to Questions

**Comments to the Author**

1. Is the manuscript technically sound, and do the data support the conclusions?

Reviewer #1: Yes

Reviewer #2: Yes

2. Has the statistical analysis been performed appropriately and rigorously? 

Reviewer #1: Yes

Reviewer #2: Yes

3. Have the authors made all data underlying the findings in their manuscript fully available?

Reviewer #1: Yes

Reviewer #2: No

4. Is the manuscript presented in an intelligible fashion and written in standard English?

Reviewer #1: Yes

Reviewer #2: Yes

5. Review Comments to the Author

Reviewer #1: The study makes a significant contribution to an under research area. The authors were thorough in their data analysis. My humble suggestion is that reference list number 6 and 7 are the same so the authors should consider deleting one and equally effect changes in the text as well

Reviewer #2: - Key words: it is better to be different from the words used in the title. It can increase the chance of finding in the searches.

- Study Setting and Site: it is too long. Please just explain the main site that you did the study and chose the participants.

- Population: the last sentence can be in the inclusion criteria part.

- Inclusion and Exclusion Criteria: educational status of caregivers is not important in your study?

- in page 6, it is mentioned that 2013 to 2021 the data requested, but again in page 7 mentioned that, between the year 2021 to 2023 was retrieved. What is the difference between this two? Make it more clear.

- "In-depth interviews were conducted using a semi-structured interview guide". This sentence repeated many times.

- "The interviews that were conducted at the facility". This sentence repeated in the third paragraph, two times in a row.

- The table 1, is a good representative of participants. It is good if you add some of this information to the inclusion criteria.

- In the discussion, it is good that you separate the various parts. But still you need more agree and disagree references to support your findings.

6. PLOS authors have the option to publish the peer review history of their article (what does this mean?). If published, this will include your full peer review and any attached files.

Reviewer #1: No

Reviewer #2: No

---

## [Author Response · Author response to Decision Letter 0]

22 Oct 2024

RESPONSE TO REVIEWERS

Reviewer 1: 

1. Reference list has been corrected and all changes made throughout the manuscript.

Reviewer 2

1. New Keywords different from words used in the title have been used.

2. Study Setting and Site has been summarized.

3. The last sentence under population was describing the entire population whiles the inclusion criteria was specific under the informal caregivers. The educational status of the caregiver did not influence the effects on the caregiver

4. The differences in the years has been corrected.

5. Repetition of “in-depth interviews” has been corrected. 

6. The ages of the children and caregivers have been added to the inclusion criteria.

7. The discussion has been separated into various parts. More references have been added.

---

## [Decision Letter · Decision Letter 1]

5 Nov 2024

Effects of Trauma-related Amputations in Children on Caregivers: An Exploratory Descriptive Study in a Developing Country

PONE-D-24-29492R1

Dear Dr. PATIENCE ACHIAMAA BARNIE,

We’re pleased to inform you that your manuscript has been judged scientifically suitable for publication and will be formally accepted for publication once it meets all outstanding technical requirements.

Kind regards,

Taher Babaee

Academic Editor

PLOS ONE

Additional Editor Comments (optional):

Reviewers' comments:

Reviewer's Responses to Questions

**Comments to the Author**

1. If the authors have adequately addressed your comments raised in a previous round of review and you feel that this manuscript is now acceptable for publication, you may indicate that here to bypass the “Comments to the Author” section, enter your conflict of interest statement in the “Confidential to Editor” section, and submit your "Accept" recommendation.

Reviewer #2: All comments have been addressed

2. Is the manuscript technically sound, and do the data support the conclusions?

Reviewer #2: Yes

3. Has the statistical analysis been performed appropriately and rigorously? 

Reviewer #2: Yes

4. Have the authors made all data underlying the findings in their manuscript fully available?

Reviewer #2: Yes

5. Is the manuscript presented in an intelligible fashion and written in standard English?

Reviewer #2: Yes

6. Review Comments to the Author

Reviewer #2: All of the comments are edited and mentioned in the text.

In addition, in inclusion criteria, still you can add some more information from the table 1. This will clarify and specify the results to the specified participants.

7. PLOS authors have the option to publish the peer review history of their article (what does this mean?). If published, this will include your full peer review and any attached files.

Reviewer #2: No

---

## [Editor Report · Acceptance letter]

24 Jan 2025

PONE-D-24-29492R1 

PLOS ONE

Dear Dr. BARNIE, 

I'm pleased to inform you that your manuscript has been deemed suitable for publication in PLOS ONE. Congratulations! Your manuscript is now being handed over to our production team.

Kind regards, 

on behalf of

Dr. Taher Babaee 

Academic Editor

PLOS ONE